# Targeted pathogen profiling of ancient feces reveals common enteric infections in the Rio Zape Valley, 725–920 CE

Drew Capone[1][iD]*, David Holcomb[2][iD], Amanda Lai[3], Tim Meade[4], Karl Reinhard[4,5], Joe Brown[2]

1 Department of Environmental and Occupational Health, School of Public Health-Bloomington, Indiana University, Bloomington, Indiana, United States of America, 2 Department of Environmental Sciences and Engineering, Gillings School of Public Health, University of North Carolina at Chapel Hill, Chapel Hill, North Carolina, United States of America, 3 Aquaya Institute, Larkspur, California, United States of America, 4 School of Natural Resources, University of Nebraska – Lincoln, Lincoln, Nebraska, United States of America, 5 Harold W. Manter Laboratory of Parasitology, University of Nebraska - Lincoln, Lincoln, Nebraska, United States of America

☯ These authors contributed equally to this work.
* dscapone@iu.edu

## Abstract

DNA analysis of ancient, desiccated feces – termed paleofeces – can unlock insights into the lives of ancient peoples, including through examination of the gut microbiome and identification of specific pathogens and parasites. We collected desiccated feces from the Cave of the Dead Children (La Cueva de Los Muertos Chiquitos) in the Rio Zape Valley in Mexico dated to 725–920 CE, for targeted pathogen analysis. First, we extracted DNA with methods previously optimized for paleofeces. Then, we applied highly sensitive modern molecular tools (i.e., PCR pre-amplification followed by multi-parallel qPCR) to assess the presence of 30 enteric pathogens and gut microbes. We detected ≥1 pathogen or gut microbe associated gene in each of the ten samples and a mean of 3.9 targets per sample. The targets detected included *Blastocystis* spp. (n = 7), atypical enteropathogenic *E. coli* (n = 7), *Enterobius vermicularis* (n = 6), *Entamoeba* spp. (n = 5), enterotoxigenic *E. coli* (n = 5), *Shigella* spp./ enteroinvasive *E. coli* (n = 3), *Giardia* spp. (n = 2), and *E. coli* O157:H7 (n = 1). The protozoan pathogens we detected (i.e., *Giardia* spp. and *Entamoeba* spp.) have been previously detected in paleofeces via enzyme-linked immunoassay (ELISA), but not via PCR. This work represents the first detection of *Blastocystis* spp. atypical enteropathogenic *E. coli,* enterotoxigenic *E. coli*, *Shigella* spp./enteroinvasive *E. coli*, and *E. coli* O157:H7 in paleofeces. These results suggest that enteric infection may have been common among the Loma San Gabriel people, who lived in the Rio Zape Valley in Mexico during this period.

**Data availability statement:** All relevant data are within the paper and its Supporting Information.

**Funding:** D.C. was supported in part by a National Institute of Environmental Health Sciences T32 Fellowship (5T32ES007018). The authors declare no competing financial interest. The funders had no role in study design, data collection and analysis, decision to publish, or preparation of the manuscript. There was no additional external funding received for this study.

**Competing interests:** The authors have declared that no competing interests exist.

## Introduction

Ancient peoples left behind a wealth of evidence, but direct biological evidence is scarce and often highly decayed. Sensitive modern molecular methods offer the opportunity to advance our knowledge of these ancient civilizations. One piece of the puzzle is desiccated human fecal material, which has been recovered at cave sites across the world [1–5]. Much can be learned from these materials; ancient stool samples, termed paleofeces, can offer insights into dietary practices, human migration, and pathogen exposures.

The ova of soil transmitted helminths (i.e., intestinal worms) have been a focus of ancient feces research for decades because they are highly persistent in the environment and are large enough to view via microscopy [6–10]. However, it is accepted in modern clinical practice that microscopy for helminth ova requires highly trained microscopists and there is potential for misclassification. The visual identification of ova in ancient feces – including *Ascaris* and hookworm – has therefore generated much debate regarding the veracity of the claims [11–17].

Modern molecular methods – including metagenomics and polymerase chain reaction (PCR) – are highly sensitive and specific [18,19]. However, nucleic acid fragments, including microbial DNA in ancient feces are highly degraded and usually present at low concentrations [20]. Given the poor analytical sensitivity associated with metagenomic methods for targets present at low concentrations, analysis is limited to communities and generally not specific pathogens without deep sequencing [21,22]. PCR is highly sensitive and specific, but specialized methods are required to maximize recovery and prevent contamination in laboratory handling [20]. Refinement of existing methods, combined with state-of-the-science molecular approaches, is increasingly being used for sensitive detection of fecal microbes in ancient feces [23,24]. Such knowledge could expand our understanding of the burden of disease among ancient peoples and pathogen transmission among different groups. Our research objective was to: (1) apply modern stool based molecular diagnostics to ancient fecal samples; and (2) assess the enteric pathogen profile of these samples.

## Methods

### Study site and population

The Cave of the Dead Children (La Cueva de los Muertos Chiquitos) in Rio Zape Valley of Mexico was excavated in 1957 and 1960 in 5X5 foot grids and 6-inch levels [25]. It exists in a low-humidity area, which slowed the degradation of the cave's contents via desiccation. Paleofeces was recovered from a midden in the back of the cave, which was composed of paleofeces, quids (i.e., masses of leaf base fiber typically expelled after chewing), cordage, diverse botanical remains and bones (Brooks *et al*. 1962: Table 1) [25]. This midden ranged from surface to 30 inches deep. It was covered by a paved activity surface, which appeared to be from the final use of the cave.

Meade previously analyzed 49 paleofeces from the cave for diverse proveniences for all types of dietary remains including pollen, seeds, fibers, and phytoliths [26]. Samples

**Table 1. Dietary observations by provenience (Grid/Depth) and Meade's 1994 laboratory analysis numbers [26].**

| Grid/Depth | Meade Lab # | Macro | Pollen |
|---|---|---|---|
| B3 0–4" | 3 | *Agave*, grass florette, | Sunflower type, Malvaceae, |
| B3 8–12" | 8 | Ground maize, *Chenopodium* seeds, sunflower achenes | Apiaceae, sunflower type |
| B3 8–12" | 8 | Juniper berries, ground maize | Maize, sunflower type, Solanaceae |
| B3 24–30" | 13 | *Agave* fiber, ground maize, *Chenopodium seeds*, *Sporobolus* caryopses, | maize |
| B3 24–30" | 13 | *Opuntia* pads, *Physalis* fruit, rodent bones | No economic pollen |
| B3 16–20" | 17 | *Agave* fiber | Brassicaceae, sunflower type |
| B4 16–20" | 1a | *Agave* fiber, sunflower achenes | Cactaceae, Caryophyllaceae, Solanaceae, sunflower type |
| B4 16–20" | 1b | Maize kernels | Maize |
| B4 16–20" | 26 | *Agave*, ground maize | No economic pollen |
| B4 20–24" | 20 | *Opuntia* pads, *Agave* fiber, maize, beans, rodent bones | No economic pollen |

were classified as human or non-human origin based on observations of these materials, as summarized by Reinhard 2017 [27]. The approach included four stages of observation, including during excavation, preliminary lab examination, rehydration and analysis of coprolite components. In 2020, we developed CoproID molecular analysis, which can be added as an additional method of inferring human origin and was applied to the Zape Cave samples [28]. All the paleofeces analyzed by Meade were consistent with human fecal material, and we randomly selected 10 samples from this collection for our analysis. In addition, the previous dietary analysis of the ten samples included in this analysis show distinct dietary data, which confirms that the samples are derived from distinct defecation events throughout the time the midden was used (Table 1) [26].

Unprocessed paleofeces from Meade's 1993 [26] study were curated for future analysis in the Pathoecology Laboratory of the School of Natural Resources, University of Nebraska-Lincoln, until 2024. Currently, they are part of the permanent Paleoparasitology Collection in the Harold W. Manter Laboratory of Parasitology, University of Nebraska State Museum.

Biogenic crystals were abundant in the 49 paleofeces analyzed by Meade [26]. Hundreds of phytoliths per gram of sample were recovered from agave, cactus, and squash. Agave phytoliths were recovered from 25 of the coprolites. Cactus and prickly pear phytoliths were present in 12 samples, and squash phytoliths were present in 5 samples. In prevalence and abundance, Agave phytoliths were overwhelmingly recovered from the coprolites. Agave was the most common food eaten at the site, and many phytoliths were liberated from the leaf bases during chewing. Agave epidermis was also found. Entire, swallowed quids were evident in some of the coprolites. Thus, the macroscopic evidence from this previous study supports the phytolith evidence that Agave was the major food at the site and that it is the main source of dietary abrasives. The macroscopic and pollen evidence of foods from the coprolites include fruits from ground cherry, maize, sunflower, pigweed and goosefoot.

Evidence of human association also came from 49 dental casts made from quids [29]. The casts from the same midden deposits as the coprolites showed a complete spectrum of age groups ranging from children to young adults and to old adults. They also show that the Zape Cave diet was abrasive. Analysis of the quids and coprolites shows that the most consistent source of dietary abrasives was from the prehistoric dietary use of Agave.

Three palaeofeces samples from the same provenience represented by the Meade paleofeces series were submitted to DirectAMS for accelerator mass spectrometry radiocarbon dating measurements [30]. The three paleofeces samples were dated to 665–770, 770–953, and 778–986 CE.

While palaeofaeces are not subject to the Native American Graves Protection and Repatriation Act (NAGPRA) or other regulations, indigenous populations with strong cultural ties to the specimens were consulted as part of previous work with these samples [30]. No permits were required for the described study, which complied with all relevant regulations.

## Nucleic acid extraction

Several precautions were taken to prevent contamination of the samples. Extractions were performed in Class IIA biological safety cabinet (BSC) that was disinfected with 10% bleach, 70% ethanol, and UV before and after use. This BSC was in a separate room from where PCR was performed. All materials were either single use items that had been purchased sterile and only opened inside the biological safety cabinet, or had been autoclaved, washed, autoclaved again, and flame sterilized before use. In addition, we purchased new bottles of reagents for this work.

We adapted nucleic acid extraction "Method B" as described in Hagan *et al*. 2020 [20], which optimized extraction methods for the recovery of DNA from paleofeces. The samples were high in fiber [29] which made them rigid and required an initial grinding step that was not previously described. After carefully breaking off small quantities of paleofeces inside a sterile whirl-pak bag (Nasco, Madison, Wisconsin), we transferred approximately 25–50 mg of paleofeces to a sterile 50mL tissue grinding tube (VWR, Radnor, Pennsylvania). The fecal material was ground into a powder by rotating the handle of the tissue grinding tube and then contents were carefully poured into a 2 mL PowerBead tube (Qiagen, Hilden, Germany) containing garnet beads. Additional material was ground and transferred until 200 mg of fecal material was achieved. We extracted 1–3 replicates of each sample depending on the quantity of fecal material available.

After loading bead tubes with paleofeces, we followed the methods described in Hagan *et al.* 2020 [20]. First, we added 400 µL of 0.5 M EDTA (VWR, Radnor, Pennsylvania), 100 µL of Proteinase K (ThermoFisher, Waltham, Massachusetts), and 750 µL of PowerBead Solution (Qiagen, Hilden, Germany) to each sample in the PowerBead tube. Then, the bead beating tube was placed inside a 50mL tube and gently rotated on a tube roller four hours at room temperature. Next, we vortexed samples for 10 minutes using a Vortex-Genie mixer (Scientific Industries, Bohemia, New York) at maximum speed and then centrifuged samples at 11,000 x g for five minutes.

Next, we carefully but firmly pushed the bottom of a Zymo-Spin V reservoir (Zymo, Irvine, California) into a MinElute column (Qiagen, Hilden, Germany) and placed the device into a sterile 50 mL centrifuge tube. We added 14 mL of Qiagen PB buffer and the entire supernatant from the bead beating tube into the chamber of the Zymo-Spin V reservoir. Then the tube was centrifuged for 4 minutes at 2000 x g, rotated 90°, and centrifuged at 2000 x g for an additional 2 minutes. Afterwards, we removed the MinElute column from the reservoir, inserted the column into a 2mL micro collection tube and centrifuged at 11,000 x g for two minutes to dry the column. Next, we pipetted 700 µL of Qiagen PE buffer (Qiagen, Hilden, Germany) into the MinElute column, centrifuged for 2 minutes at 11,000 x g, discarded the flow-through and then repeated this step with fresh PE buffer. Finally, we added 30 µL of Qiagen EB buffer (Qiagen, Hilden, Germany) to the MinElute column, incubated at room temperature for 5 minutes, centrifuged at 11,000 x g, retained the flow-through, and repeated this step. This process resulted in 60 µL of template, which was stored at 4°C for less than 24 hours and then stored at −80°C. The average concentration of dsDNA in our extracts – measured via a Qubit dsDNA Assay Kit on a Qubit 4 Fluorometer (ThermoFisher, Waltham, Massachusetts) – was 5.1 nanograms per microliter (range: 0.30–19.6).

On each day of extractions, we included one negative extraction control to monitor for contamination. We did not spike in extraction control material to the samples to limit the potential for contamination.

## Pre-amplification

We used the TaqMan PreAmp Master Mix Kit to pre-amplify our target sequences and lower our limit of detection (ThermoFisher, Waltham, Massachusetts). First, we pooled the 38 forward and 38 reverse primers (IDT, Coralville, Iowa) such that concentration of each primer was 0.18 µM, which equates to the final concentration required by the kit. Then we prepared 20 µL pre-amplification PCR reactions for each sample according to the manufacturer instructions, which included TaqMan PreAmp Mastermix (10 µL), the diluted primer pool (5 µL), DNA template (5 µL). The resulting reaction was run on a Bio-Rad CFX 96 Touch thermocycler (Bio-Rad, Hercules, CA) with an initial activation at 95°C for 10 minutes, followed by 14 cycles of 95°C for 15 seconds followed by 60°C for 4 minutes.

## Real-time PCR

We developed a custom TaqMan Array Card (TAC) according to Liu *et al*. 2013 [31] and 2016 [32] (Table S1, Table S2). The pathogens and gut microbes we assessed included helminths (*Ancylostoma duodenale*, *Ascaris lumbricoides*, *Enterobius vermicularis*, *Hymenolepis nana*, *Necator americanus*, *Strongyloides stercolaris*, and *Trichuris trichiura*), protozoa (*Acanthamoeba* spp., *Balantidium coli*, *Blastocystis* spp., *Cystoisospora belli*, *Cyclospora cayetanensi*, *Cryptosporidium* spp., *Enterocytozoon bieneusi*, *Encephalitozoon intestinalis*, *Entamoeba hystolytica*, *Entamoeba* spp., *Giardia* spp.), and bacteria (*Campylobacter jejuni/coli*, *Clostridium difficile*, *E. coli* O157:H7, enteroaggregative *E. coli*, enteropathogenic *E. coli*, enterotoxigenic *E. coli*, *Helicobacter pylori*, *Shigella*/enteroinvasive *E. coli*, *Plesiomonas shigelloides*, *Salmonella* spp., shiga-toxin producing *E. coli*, *Yersinia enterocolitica*). The assays used were previously validated on modern fecal material with 100% sensitivity and 95%−100% specificity [32]. The average amplicon length for an assay was 116 bp (minimum = 54 bp, maximum = 238 bp) (S2 Table). In addition, the card included assays for enteric 16S rRNA and phocine herpes virus (PhHPV) [32]. The enteric 16S rRNA assay –described in Rousselon *et al*. 2004 [33] – was designed to detect a cluster of phylotypes, called Fec1, corresponding to 5% of the human fecal microflora [32].

The TAC was prepared by combining 6.67 µL of pre-amplification product [34], with 31.3 µL of molecular grade water, 2 µL of a synthetic DNA sequence matching the PhHPV assay to monitor inhibition ($10^6$ copies per µL), and 60 µL of AgPath-ID™ One-Step RT-PCR Reagents (Applied Biosystems, Waltham, MA). We used the standard TAC cycling conditions with a 1°C/s ramp rate between all steps: 45°C for 20 minutes, 95°C for 10 minutes, then 45 cycles of 95°C for 15 seconds and 60°C for 1 minute [31,32]. The TAC performance was evaluated using an 8-fold dilution series ($10^9$-$10^2$ gene copies per reaction) of an engineered combined positive control that was developed using methods from Kodani and Winchell 2012 [35]. The linearity and efficiency the targets were within normative standards (linearity: 0.97-1.0, efficiency: 87%−102%) (S2 Table). Each day of TAC analysis, one PCR positive control and one negative extraction control were analyzed. Quantification cycle (Cq) values were determined by manual thresholding and included comparison of each assay's fluorescent signal against the daily negative and positive controls (S1 Fig). Any target that amplified past a Cq of 40 was categorized as negative to reduce the potential for false positives [32]. The theoretical limit of detection – a result of the dilutions used – was 60 gene copies per gram solids (S3 Table).

## Digital PCR

We performed digital PCR with a QIAcuity 4 instrument (Qiagen, Hilden, Germany) to assess the presence of human mitochondrial (mtDNA) DNA in the paleofeces samples to assess if fecal material was of human origin (S4 Table) [36]. Human mtDNA is present in human feces because intestinal epithelial cells and leucocytes are shed from the intestinal lining into fecal matter. This assay has demonstrated high sensitivity (100%) and specificity (97%) to fresh human feces. We prepared reactions with QIAcuity Probe Mastermix, 200 nM forward and reverse primers, 800 nM probe, and 2 µL of raw template (i.e., no pre-amplification was performed). Thermocycling conditions were 95°C for two minutes, followed by 45 cycles of 95°C for 15 seconds and 55°C for 60 seconds. We included one positive and one negative control on each dPCR nanoplate. We differentiated positive and negative partitions by manual thresholding between the bands of the positive and negative controls. Samples with less than three positive partitions were classified as negative to reduce the potential for false positives.

## Results

### Controls

Assays for the four PCR positive controls exhibited positive amplification as expected (Cq ~ 18). There was no positive amplification for any target in the four negative extraction controls, except for the 16S rRNA assay. Microbial DNA contamination of Taq polymerase in PCR mastermix has been documented [37,38] including for the 16S rRNA gene [39,40].

According to the quality control of the manufacturer, the 20 μL pre-amplification reactions contained ≤2 copies of the 16S rRNA gene [40]. The pooled primers in the reaction included primers for the 16S assay, which if present, may have amplified this contamination. The spiked inhibition control – which was a synthetic DNA sequence – amplified consistently (Cq~20) for all samples.

## Molecular Results

We detected ≥1 enteric pathogen or gut microbe in each of the ten paleofeces from Mexico and a mean of 3.9 targets per sample out of the 30 assessed (Table 2, S5 Table). The targets detected in the ten samples included *Blastocystis* spp. (n = 7), atypical enteropathogenic *E. coli* (n = 7), *Enterobius vermicularis* (n = 6), *Entamoeba* spp. (n = 5), enterotoxigenic *E. coli* (n = 5), *Shigella* spp./enteroinvasive *E. coli* (n = 3), *Giardia* spp. (n = 2), and *E. coli* O157:H7 (n = 1). One sample was positive for human mtDNA via dPCR at a high concentration (approximately $10^5$ gene copies/gram paleofeces) and was only positive for *Blastocystis* spp. While, human mtDNA was not detected in the remaining nine samples, all 10 samples were positive for the enteric 16S rRNA target.

We analyzed replicates from nine of the ten samples. This included six samples in duplicate and three samples in triplicate, which was limited by the mass of material available. Among these nine samples, there were 40 instances where a pathogen associated gene was detected in a sample and could be compared against detection in the other replicate samples. We observed perfect concordance (i.e., both duplicates or all three triplicates positive) among 60% (n = 24/40) of replicates, moderate concordance (two out of three triplicates positive) among 5.0% (n = 2/40), and poor concordance (one detection out of two or three replicates) among 35% (n = 14/40).

## Discussion

We detected diverse enteric pathogens in 1,100−1,300 year-old paleofeces from Mexico. The 60% prevalence of pinworm (i.e., *Enterobius vermicularis*) we observed is greater than the 34% (n = 34/100) prevalence determined via microscopy on

**Table 2. Prevalence of molecular targets.**

| Target | Prevalence | N (out of 10) |
|---|---|---|
| enteric 16S rRNA[†] | 100% | 10 |
| *Blastocystis* spp | 70% | 7 |
| Enteropathogenic *E. coli* (atypical) | 70% | 7 |
| *Enterobius vermicularis* (pinworm) | 60% | 6 |
| *Entamoeba* spp. | 50% | 5 |
| Enterotoxigenic *E. coli* | 50% | 5 |
| Enteropathogenic *E. coli* (typical) | 30% | 3 |
| *Shigella*/EIEC | 30% | 3 |
| *Giardia* spp. | 20% | 2 |
| *E. coli* O157:H7 | 10% | 1 |
| human mtDNA | 10% | 1* |

Note: The following were not detected: *Acanthamoeba* spp., *Ancylostoma duodenale*.

*Ascaris lumbricoides*, *Blantidium coli*, *Cystoisospora belli*, *Cyclospora cayetanensi*, *Campylobacter jejuni* & *coli*, *Clostridioides difficile* B, *Cryptosporidium* spp., *Enterocytozoon bieneusi*, *Encephalitozoon intestinalis*, *Entamoeba histolytica*, *Hymenolepis nana*, *Helicobacter pylori*, *Necator americanus*, *Plesiomonas shigelloides*, *Salmonella* spp., *Strongyloides stercolaris*, *Trichuris trichiura*, *Yersinia enterocolitica*, Enteroaggregative *E. coli*, Shiga-toxin producing *E. coli*. EIEC = Enteroinvasive *E. coli*.

*The positive sample exhibited a strong positive signal.

[†]This assay was designed for human feces, but may cross react with some animal feces [33].

paleofeces from the same cave in Mexico. PCR has greater sensitivity than microscopy, but our small sample size suggests cautious interpretation. Protozoan pathogens we detected, including *Giardia* spp. [5] and *Entamoeba* spp. [41] have been previously detected in paleofeces via enzyme-linked immunoassay (ELISA), but have not via PCR. In addition, this work represents the first detection of *Blastocystis* spp., atypical enteropathogenic *E. coli,* enterotoxigenic *E. coli*, *Shigella* spp./enteroinvasive *E. coli*, and *E. coli* O157:H7 in paleofeces.

There is debate whether helminths other than pinworm were circulating among people in the Americas before the Columbian exchange began in 1492 [11]. The molecular detection of pinworm supports the use of qPCR assays, combined with microscopy and sequencing, to better understand which helminths were circulating in the Americas prior to the Columbian Exchange [23,24]. Given the potential for ova degradation, reported visual observations of hookworm and *Ascaris* ova in paleofeces from the Americas pre-1492 would be strengthened by further molecular validation. However, definitive conclusions should not be drawn from a small sample size of paleofeces from a single cave.

Several of the non-detect results contrast with previous findings. However, a non-detect does not indicate that the target was not initially present in the sample. It does indicate that we were unable to detect it with the methods used. For example, 61% (55/90) of paleofeces from the same cave in Mexico exhibited a strong positive signal for *Cryptosporidium parvum* via ELISA [42]. There are several possible explanations for this discrepancy. It is possible that the *Cryptosporidium* antigens measured had greater environmental persistence than the nucleic acids inside the oocysts. Second, our analysis was limited to ten samples. Analysis of other paleofeces samples from this cave may have detected *Cryptosporidium*. This discrepancy further highlights the importance of using multiple complementary assays when interpreting evidence from ancient fecal material.

Cautious interpretation is also warranted for the infrequent detection of human mtDNA. These paleofeces were previously identified as human based on dietary materials, morphology and size [1,42], but such analysis may be subjective. The presence of *Enterobius vermicularis* in most samples also provides strong evidence of human origin. *Enterobius vermicularis* is a human specific pathogen, though it can infect some species of apes [43–45]. The detection of human mtDNA at a high concentration from one sample – and the detection of the enteric 16S rRNA assay in all samples – provides additional confidence that at least some of these paleofeces are from humans. Although, we did not test for mtDNA from animals.

Human mtDNA is often found at lower concentrations in human feces than genes from other microbial organisms [36,46] and its persistence in the environment, relative to the pathogenic genes we detected, is unclear. In our previous work with modern fecal material from school-age children, we found the concentration of enteric 16S rRNA was approximately 9,300 times greater than human mtDNA [47]. If these signals persisted at the same or similar rates, then this initial difference in concentration may explain why enteric 16S rRNA was observed above our limit of detection in each sample but human mtDNA was only detected in one sample.

If some or all these feces are from humans, then this study suggests poor sanitation among the Loma San Gabriel culture from 600-800 CE resulted in exposures to fecal wastes in the environment. Human and animal feces may contain enteric pathogens, which are transmitted via drinking water, soils, food, flies, and fomites [48,49]. Some of the pathogens we detected are zoonotic, meaning that they can be shed by animals as well as humans. For example, *Shigella* spp. is considered specific to humans, though the sample positive for *Shigella* spp. was negative for human mtDNA [50].

We used PCR to detect the presence or absence of genes that are found in specific microorganisms [19]. PCR requires prior knowledge of the target sequence, which limited our analysis to specific pathogen and gut microbe associated genes. Our pre-amplification and PCR methods were also limited to the specific target of interest and we were unable to provide information about other DNA sequences that may have been present. Metagenomics is an alternative method that involves sequencing all the genetic material in sample [18]. It provides the genetic code of the entire microbial community, including reads for known and unknown organisms. Metagenomic methods have been used to characterize microbial communities and reconstruct ancient microbial genomes from paleofeces

[30,51,52]. Sequencing methods, however, have limitations in identifying microbial DNA at the genus or species level as we have done here. Metagenomics has a substantially higher limit of detection than PCR, and sequencing may miss low abundance genomes [21,22]. If genomes are similar, then bioinformatics pipelines may be unable to resolve species or genus-level details. Accurate identification also relies on the quality of reference databases. If these databases include misidentified genomes, or if specific genomes are not well represented in the database, then it may be challenging to accurately identify the aligned reads.

There are several limitations associated with this work. First, we analyzed a small number of samples from a single cave. An analysis of additional samples may have detected other pathogen-associated genes or a different prevalence of the genes we detected. We observed perfect concordance between replicates for most targets that we detected. Intra-stool heterogeneity has been demonstrated for enteric pathogen in modern feces [53]. We did not homogenize entire ancient fecal samples, which is common when working with fresh feces, and may have contributed to the heterogeneity in detection. The persistence of the gene targets in paleofeces is not well characterized. Evidently some pathogen associated genes persisted in the samples for 1,100−1,300 years, but it is unclear if other genes may have decayed beyond our ability to detect them. Damage and chemical modifications in ancient DNA—such as cytosine deamination—can be assessed through sequencing and bioinformatic analysis to help distinguish endogenous ancient DNA from modern contamination [54]. Uracil-DNA Glycosylase (UDG) treatment has been developed to remove uracil residues and repair resulting abasic sites [55]. While we did not employ these methods, they are primarily used in metagenomic studies and are less applicable to qPCR-based detection. However, the presence of deaminated nucleotides in ancient feces may have impaired primer and probe binding or amplification efficiency, potentially leading to underestimation of the prevalence of our target genes.

PCR inhibitors may have unique impacts on different PCR assays, and assay specific inhibition may have occurred given the large targets assays we used [56]. While our extraction methods had been optimized for paleofeces, we prioritized recovery of small DNA fragments by, in part, omitting inhibitor removal steps that would typically be used in fresh stool extractions [20]. Further, we did not conduct molecular analyses in a clean room dedicated to ancient DNA analysis [57]. While we worked under sterilized BSL2 conditions with sterile single use reagents and observed no contamination in our negative controls, we cannot completely rule out the potential for cross contamination. We did not perform UV surface sterilization of the paleofeces, as enteric pathogens replicate exclusively in the gut and are not part of endogenous soil microbiota. Potential surrounding soil contamination during excavation was highly unlikely to introduce these organisms to paleofeces surfaces. Finally, there is always a possibility of nonspecific amplification in PCR reactions, and if nonspecific amplification occurred, we may have overestimated the prevalence of the targets in our samples. However, we did not observe nonspecific amplification in our negative controls in this study or a previous study that used the same assays on a larger number of samples [47].

We detected genes associated with eight enteric pathogens and one common intestinal microbe (i.e., *Blastocystis*), many of which have never been detected before in paleofeces. These results indicate that modern molecular techniques are an effective tool to screen paleofeces – and potentially other ancient samples – for multiple gene-based targets of interest. The application of these methods to other ancient samples offers the potential to expand our understanding of how ancient peoples lived and the pathogens that may have impacted their health.

## Supporting information

**S1 Table.  qPCR assays.**
(DOCX)

**S2 Table.  qPCR QA/QC.**
(DOCX)

**S1 Fig. Amplification curves.**
(DOCX)

**S3 Table. MIQE Checklist.**
(DOCX)

**S4 Table. dPCR assay.**
(DOCX)

**S5 Table. Supplemental data file.**
(XLSX)

## Author contributions

**Conceptualization:** Drew Capone, David Holcomb, Karl Reinhard, Joe Brown.

**Data curation:** Tim Meade.

**Formal analysis:** Tim Meade.

**Investigation:** Drew Capone, David Holcomb, Amanda Lai, Karl Reinhard.

**Methodology:** Drew Capone, David Holcomb, Amanda Lai, Karl Reinhard, Joe Brown.

**Project administration:** Karl Reinhard, Joe Brown.

**Resources:** Karl Reinhard.

**Writing – original draft:** Drew Capone.

**Writing – review & editing:** David Holcomb, Amanda Lai, Karl Reinhard, Joe Brown.

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
