## [Decision Letter · Decision Letter 0]

10 Mar 2025

Dear Dr. Capone,

Thank you for submitting your manuscript to PLOS ONE. After careful consideration, we feel that it has merit but does not fully meet PLOS ONE’s publication criteria as it currently stands. Therefore, we invite you to submit a revised version of the manuscript that addresses the points raised during the review process.

We look forward to receiving your revised manuscript.

Kind regards,

Elham Kazemirad, Ph.D

Academic Editor

PLOS ONE

Journal Requirements:

2. In your manuscript, please provide additional information regarding the specimens used in your study. Ensure that you have reported human remain specimen numbers and complete repository information, including museum name and geographic location.

For more information on PLOS ONE's requirements for paleontology and archeology research, see https://journals.plos.org/plosone/s/submission-guidelines#loc-paleontology-and-archaeology-research .

4. Thank you for stating the following financial disclosure: [D.C. was supported in part by an NIH T32 Fellowship (5T32ES007018-44). The authors declare no competing financial interest.]. 

5. Thank you for stating in your Funding Statement: [D.C. was supported in part by an NIH T32 Fellowship (5T32ES007018-44). The authors declare no competing financial interest.].

6. We note that there is identifying data in the Supporting Information file < Supplemental_v2b.docx>. Due to the inclusion of these potentially identifying data, we have removed this file from your file inventory. Prior to sharing human research participant data, authors should consult with an ethics committee to ensure data are shared in accordance with participant consent and all applicable local laws.

-Location data

Please remove or anonymize all personal information (Name) ensure that the data shared are in accordance with participant consent, and re-upload a fully anonymized data set. Please note that spreadsheet columns with personal information must be removed and not hidden as all hidden columns will appear in the published file.

7.  Please include captions for your Supporting Information files at the end of your manuscript, and update any in-text citations to match accordingly. Please see our Supporting Information guidelines for more information: http://journals.plos.org/plosone/s/supporting-information .

Reviewers' comments:

Reviewer's Responses to Questions

**Comments to the Author**

1. Is the manuscript technically sound, and do the data support the conclusions?

Reviewer #1: Partly

Reviewer #2: Yes

Reviewer #3: Partly

Reviewer #4: Yes

2. Has the statistical analysis been performed appropriately and rigorously?

Reviewer #1: N/A

Reviewer #2: N/A

Reviewer #3: No

Reviewer #4: Yes

3. Have the authors made all data underlying the findings in their manuscript fully available?

Reviewer #1: Yes

Reviewer #2: Yes

Reviewer #3: No

Reviewer #4: No

4. Is the manuscript presented in an intelligible fashion and written in standard English?

Reviewer #1: Yes

Reviewer #2: Yes

Reviewer #3: No

Reviewer #4: Yes

Reviewer #1: Many thanks to the editor for asking me to review this paper, and many thanks to the authors for their contribution to the field.

Here are my comments. Obviously I may have unwillingly distorted the points made by the authors and I’m fully open to be contradicted and shown to be wrong.

L 27-28 : it is true that modern molecular tools such as PCR can be used in this kind of studies, but this is known since several decades now. So I’d tend to put the final emphasis of your abstract on something else you effectively demonstrated in your paper rather than this.

L. 42 : Note that molecular identification of ancient pathogens or other past organisms has also, and still does, generated much debate. I’d suggest diminishing the idea of opposed methods and preferring the idea of proxy complementarity. Many studies showed how looking in parallel for molecular signatures, micro and macro remains, where actually much more informative rather than only aDNA or microscopy for instance.

L. 49-51 : once again, studying ancient pathogens from coprolites through their molecular signature is not something new in 2025 but your wording may suggest the contrary. See for example : Appelt S, Armougom F, Le Bailly M, Robert C, Drancourt M. Polyphasic analysis of a middle ages coprolite microbiota, Belgium. PLoS One. 2014 Feb 28;9(2):e88376. doi: 10.1371/journal.pone.0088376. PMID: 24586319; PMCID: PMC3938422. And it included microscopy-based diagnosis.

L. 51 : could you be more specific on how such knowledge could expand our understanding of how peoples lived ?

L. 61 : as you are not solely speaking to molecular biologists and microbiologists I think you should be more accurate on site description and chronological methods, even if it was previously published. You are also speaking to paleo/archaeo scientists in this paper. Note that you mentioned “previously collected […] material” (l. 57), a naive reader would then certainly understand that there is previous studies of these materials. You certainly should cite them here too.

L.66-71 Could you please specify if this MSC is located in a clean room dedicated to aDNA analysis, or if it is at least dedicated to aDNA samples (meaning, it never saw any modern feces sample)?

L.118-130 : Could you tell more in the core paper about these targets? How long are their amplicons, were they previously tested on modern samples? I think these informations should be made clear in the core text without having to dive in the supplementary files.

L. 148-159: Wouldn’t it be appropriate to also test non-human targets?

Is there any control performed with the surrounding sediment to distinguish the true feces’ content from surrounding leaching and environmental agents?

L. 201-203 : Did you have the opportunity to test the exact same coprolites (in order to know if some of them were positive under the microscope but not after PCR).

L. 213-216 : this also pleads for mentioning multi-proxy approaches I mentioned before. Far from diminishing your own approach, as you are potentially able to highlight it.

L. 218-212 : Aside from the debate in itself, I would highly recommend to chose another wording for this sentence, as the detection of E. vermicularis per se does not support at all the absence of other helminths in pre-contact America. You are here substantially saying that most, if not all, published detections of Ascaris and hookworms from pre-contact America are misidentification. It’s a strong statement that should be based on case by case counter-identification based on published pictures, and not derived from the molecular detection of E. vermicularis within these 10 samples (I’m not working on this region/period, so don’t worry about me being triggered here).

L. 227 : no other animals were tested if I’m correct (like, most commonly encountered cave dwelling mammals then and now).

L. 229 : but you did not mention any test performed with sediment samples from the sedimentary layers surrounding these coprolites.

L. 233 : if most of these pathogens are shared between human and non-human hosts, then how could you conclude mostly about human behavior and waste management? I know there is a whole published archaeological background that could be used here to support your point, but you are not using it. So a naive reader can only think you decided a little bit arbitrarily it was human rather than non-human coprolites and pathogens, to document human waste management.

L. 235: Did Shigella and human mtDNA detection match the same coprolite?

L. 252-261 : You did not provide any sequencing set that could be compared with environmental samples and assessed for aDNA damage patterns. Wouldn’t be considered as limitations?

As a whole I think this paper is interesting when suggesting that a well documented qPCR assay for modern samples and targeting a large variety of enteric pathogens could be used in paleodiagnosis. But as it is written at least, it is not at all integrated in archaeo/paleosciences. While it seems to highlight molecular and PCR based approaches as something new, it does not follow usual standards of authenticity in paleomicrobiology (typically providing sequence data showing damage patterns and control samples from the surrounding sedimentary layers, or at least discussions about depositional and post depositional process). It does not introduce at all the archaeological and sampling context, conditions and methods, and barely mentions previous studies from the same site. Finally, it tries to address the biological origin of the samples, but did not mention any real attempt to falsify the human origin hypothesis. It leaves me with the feeling that the paper, though technically interesting, did not, even remotely, attempt to address the archaeo/paleoscientific questions it was supposed to.

Reviewer #2: The authors present a study of 10 paleofeces from Mexico. They used a pre-amplification method followed by a qPCR analysis and a dPCR analysis to examine enteric pathogens and human mtDNA respectively. The manuscript was easy to read and follow. I am conviced that the authors have detected ancient enteric pathogens, which is of scientific interest. However, I have a few concerns I would like to see addressed:

1. The authors address that they do not use metagenomic analysis, even though that is the norm in the ancient DNA field. Their arguments against it should be backed up by more data. For example in lines 246 and 247, the authors claim "Metagenomics

has a higher limit of detection than PCR, and sequencing may miss low abundance genomes" but do not provide a citation for this statement. I would be surprised if this were true. Especially as the authors do not discuss a large bais they will have when using PCR primers: ancient DNA is very short and most fragments will not be amplified using primers so they are able to see only the very well preserved, long DNA fragments using this method. I would rather say the strength of this method could be cost, which is important. NGS library preperation and sequencing to levels deep enough to detect these microbes would be far more expensive than this method.

2. I would like a discussion of how the TaqMan kit they use could cause biases. This kit is designed to be used for cDNA and not genomic DNA and could thus cause biases due to non-specific amplification.

3. My biggest concern is the human mtDNA. Without sequencing it is impossible to claim that the human mtDNA comes from a true ancient source. In my experience with ancient DNA, samples themselves are often contaminated with modern human DNA. Thus a negative extraction control does not indicate that there is no potential modern human contamination. One would need to sequence this DNA and show patterns of ancient damage to be convincing that this comes from a human source. I think that making NGS libraries and sequencing are outside the scope of this study, however I would like this potential modern contamination discussed.

4. My last comment is minor: There is a typo in line 74 "The samples from were high in fiber [22]". I assume a word is missing?

Reviewer #3: Sequencing of Samples: The manuscript presents interesting findings, but an essential issue is the lack of sequencing for all amplified products. It is necessary to sequence all samples to confirm that the amplification is specific and accurately represents the target organisms. Without sequencing, there is a significant risk of false positives, particularly when working with microbial eukaryotes.

Monitoring Evolution Over Time: The study could be greatly strengthened by incorporating a temporal analysis of the evolution of these organisms. This would add a valuable dimension to the research, providing insights into changes in pathogen profiles over time and improving the impact of the study.

Language and Terminology: The manuscript would benefit from a thorough review of the language for clarity and readability. There are typographical errors that should be corrected. Additionally, statements such as "Blastocystis is a pathogen" should be carefully reconsidered. The pathogenicity of Blastocystis remains debated, and it would be more appropriate to refer to it as an organism commonly found in the gut rather than labeling it as a definitive pathogen.

Minor suggestions:

Clarification of Pathogenicity Terminology: The manuscript states that Blastocystis is a pathogen. Given the ongoing debate regarding its pathogenicity, the authors should revise this statement to reflect current scientific consensus, such as referring to it as a "commonly detected gut microbe" rather than a definitive pathogen.

Grammar, Spelling, and Typographical Errors: The manuscript contains typographical and grammatical issues that should be corrected. A careful proofreading is necessary to improve readability and clarity.

Consistency in Terminology: Ensure that terminology such as paleofeces vs. coprolites is used consistently throughout the manuscript to avoid confusion.

PCR and qPCR Methodological Justification: The authors should provide a stronger justification for their choice of PCR/qPCR as the primary detection method without sequencing. If sequencing is not included, they should explicitly discuss the limitations and potential risks of false positives.

Table Formatting and Data Presentation:

Table 1: Ensure uniform formatting of columns and clearly indicate sample sizes for each prevalence calculation.

Consider adding confidence intervals where applicable for prevalence values.

Figures and Legends:

Figures should have more descriptive legends to ensure they are self-explanatory.

If possible, improve resolution and clarity of any images, particularly those displaying results.

Reference Updates: Ensure all references are up-to-date and correctly formatted according to PLOS ONE guidelines.

Statistical Interpretation:

Where applicable, clarify the statistical significance of the findings.

If sample sizes are small, acknowledge this limitation explicitly.

Reviewer #4: This study applies modern molecular techniques to analyze ancient desiccated feces (paleofeces) collected from caves in the Rio Zape Valley, Mexico (725–920 CE). Paleofeces can offer invaluable insights into ancient dietary practices, human migration, and pathogen exposure. The authors leverage PCR pre-amplification followed by multi-parallel qPCR to detect the presence of 30 enteric pathogens, a substantial advancement over traditional methods like microscopy, which has been used for identifying soil-transmitted helminths but carries a risk of misclassification due to observer bias. The study provides important insights, as it represents the first molecular detection of several pathogens in paleofeces, including Blastocystis spp., atypical enteropathogenic E. coli, enterotoxigenic E. coli, Shigella spp., and E. coli O157:H7. These findings demonstrate the power of modern molecular tools like PCR to detect degraded ancient DNA and highlight how PCR can be applied to study pathogens in ancient biological materials.

However, there are several critical technical aspects that need more clarification to ensure reproducibility and robustness in the methodology:

1. Origin and Dating of Samples:

o The authors do not specify how the dating of the samples was performed. Detailing this information would strengthen the study's credibility and accuracy in placing these samples in their correct historical context.

2. DNA Extraction:

o The study does not mention the use of a dedicated clean room for ancient DNA (aDNA) extraction. Ancient DNA is highly degraded and prone to contamination from modern DNA. Best practices for aDNA extraction require a physically separated clean lab with HEPA-filtered positive-pressure air and full-body PPE (including face shields, lab coats, and gloves). The study should clarify whether these practices were implemented.

o Additionally, Uracil-DNA Glycosylase (UDG) treatment, which helps correct DNA damage from deamination, is not mentioned. If UDG was not used, the authors should acknowledge this limitation in their study. If it was used, the specific procedure should be described.

o The authors should also provide more information about DNA purity (e.g., A260/A280 and A260/A230 ratios) and whether they assessed the quality and fragmentation of the DNA (e.g., using a Bioanalyzer or TapeStation). Without assessing DNA integrity, there is a risk that the DNA may be too degraded to amplify efficiently, potentially affecting the results.

3. qPCR Methodology:

o The study mentions using 38 forward and 38 reverse primers from IDT (Coralville, Iowa), but the selection and validation of these primers for use with degraded or ancient DNA are not described. The authors should explain if these primers were specifically tested for their ability to work with low-abundance or fragmented DNA. It would be helpful to include a discussion of how they ensured the primers' efficacy in amplifying ancient DNA sequences.

4. Digital PCR:

o The use of digital PCR (dPCR) to detect human mitochondrial DNA (mtDNA) in paleofeces is a sound approach to confirm the human origin of the samples. However, modern human mtDNA contamination is a common issue in laboratories, as it is highly abundant. The authors should address the potential for contamination and explain how they mitigated the risk of false positives for human fecal origin. This could be crucial in ensuring the accuracy of their findings regarding the genuine ancient DNA sequences.

Results:

Did the data retrieved from this study deposited in online repositories?

The study presents a methodologically strong approach, combining modern molecular tools like PCR and dPCR with innovative techniques for detecting pathogens in ancient feces. However, there are critical technical gaps, particularly regarding cleanroom practices, DNA extraction protocols, and primer validation for ancient DNA. If these precautions were not followed, the results might be subject to contamination or bias. I recommend major revisions to address these concerns and strengthen the overall reliability and reproducibility of the methodology.

**Do you want your identity to be public for this peer review?** For information about this choice, including consent withdrawal, please see our Privacy Policy

Reviewer #1: No

Reviewer #2: No

Reviewer #3: No

Reviewer #4: No

---

## [Author Response · Author response to Decision Letter 1]

6 Jun 2025

We appreicate the author's time and thoughtfulness. We feel the feedback has greatly strengthened the manuscript. Please see the attached document for our point by point response to comments.

---

## [Decision Letter · Decision Letter 1]

9 Jul 2025

Dear Dr. Capone,

Thank you for submitting your manuscript to PLOS ONE. After careful consideration, we feel that it has merit but does not fully meet PLOS ONE’s publication criteria as it currently stands. Therefore, we invite you to submit a revised version of the manuscript that addresses the points raised during the review process.

We look forward to receiving your revised manuscript.

Kind regards,

Elham Kazemirad, Ph.D

Academic Editor

PLOS ONE

**Journal Requirements:**

Reviewers' comments:

Reviewer's Responses to Questions

**Comments to the Author**

Reviewer #1: All comments have been addressed

Reviewer #2: All comments have been addressed

Reviewer #3: (No Response)

Reviewer #4: (No Response)

2. Is the manuscript technically sound, and do the data support the conclusions?

Reviewer #1: Yes

Reviewer #2: Yes

Reviewer #3: No

Reviewer #4: (No Response)

3. Has the statistical analysis been performed appropriately and rigorously?

Reviewer #1: N/A

Reviewer #2: N/A

Reviewer #3: No

Reviewer #4: (No Response)

4. Have the authors made all data underlying the findings in their manuscript fully available?

Reviewer #1: Yes

Reviewer #2: Yes

Reviewer #3: Yes

Reviewer #4: (No Response)

5. Is the manuscript presented in an intelligible fashion and written in standard English?

Reviewer #1: Yes

Reviewer #2: Yes

Reviewer #3: Yes

Reviewer #4: (No Response)

**Reviewer #1:**  I have read the authors' responses to my comments. While the article still presents some of the weaknesses previously identified by myself and the other reviewers, I acknowledge that the authors have addressed these points honestly and have clearly acknowledged and discussed the limitations. In light of this transparent and constructive approach, I believe the article is suitable for publication.

**Reviewer #2: ** The authors have adequatly addressed my concerns and I believe the manuscript is now ready for publication.

**Reviewer #3:**  Thank you for your revision and responses. I appreciate the efforts made to improve the manuscript. However, after reviewing the revised version and the point-by-point response, I remain unconvinced that my concerns have been adequately addressed. Below, I outline the key issues that persist:

1. Sample Size and Statistical Justification

While I understand the constraints of working with ancient material, the manuscript still lacks any form of quantitative justification for the sample size (n = 10), nor does it provide a clear explanation of the implications of such a limited dataset. There is no power analysis or alternative rationale. This omission weakens the strength of the claims, especially those that discuss prevalence or make broader epidemiological inferences. I strongly recommend softening these conclusions and clearly communicating the exploratory nature of the work.

2. Overinterpretation of Results

Several conclusions—such as the suggestion that "enteric infections were common" among the Loma San Gabriel people—are not sufficiently supported given the data. This interpretation does not take into account:

the potential for false positives, especially with low-level amplification,

lack of authentication for ancient DNA (e.g., no sequencing or damage pattern analysis),

limited replication and variability in concordance between replicates,

and absence of confirmatory tests (e.g., amplicon sequencing or histological correlation).

These concerns are not fully addressed in the revised manuscript.

3. Limited Replication and Concordance

The replication data show that 35% of targets were detected in only one of two or three replicates. This level of variability raises questions about reproducibility and the robustness of detections. While some variability can be expected in low-biomass, degraded samples, the extent observed here undermines confidence in the findings. This needs to be more transparently acknowledged in the main text and considered in the interpretation of prevalence data.

4. Molecular Authentication of Ancient DNA

The authors did not incorporate any ancient DNA authentication strategies—such as damage pattern analysis, uracil-DNA glycosylase treatment, or independent validation through sequencing—to demonstrate that the detected targets represent ancient rather than modern or environmental contamination. Given that the study’s novelty hinges on being the “first” to report certain pathogens in paleofeces, this omission is significant and weakens confidence in the findings.

5. Insufficient Biological Context for Detected Taxa

The manuscript continues to list pathogen-associated genes without offering substantive biological interpretation. While additional references were added, the discussion remains superficial. For example, the ecological relevance, potential zoonotic transmission, or expected prevalence in ancient vs. modern contexts are not explored for many of the bacteria or protozoa. Without such context, the findings are descriptive rather than explanatory.

6. Wording and Clarity

Despite some edits, the manuscript continues to rely on vague or subjective language (e.g., “diverse,” “common,” “novel”). I recommend further revision to use precise, evidence-based language. The interpretation of results should be framed with appropriate caution given the study’s limitations.

7. Controls and Contamination

The authors note that 16S rRNA genes were detected in negative controls and attribute this to contamination of commercial reagents. While this is plausible and a known issue, it underscores the need for independent validation (e.g., sequencing amplicons or removing affected targets from interpretation). This issue is not sufficiently addressed in the discussion.

8. Conclusions

The authors conclude that modern molecular techniques are effective for detecting pathogens in paleofeces. While this may be true in principle, the methods applied here lack several key elements (e.g., authentication, replication consistency, host confirmation in most samples) that are critical in ancient DNA studies. The paper would benefit from reframing this conclusion as a proof-of-concept rather than an authoritative epidemiological report.

Summary

In its current form, this manuscript presents technically interesting data but overstates its conclusions. Additional caution in interpretation, further discussion of limitations, and ideally, more robust molecular validation or justification would be needed for the study to meet publication standards. I recommend major revision before further consideration.

**Reviewer #4: ** - Use of Class IIA Biological Safety Cabinet Instead of a Clean Room

DNA extractions were performed in a Class IIA BSC, disinfected with 10% bleach, 70% ethanol, and UV light, rather than a dedicated aDNA clean room. Class IIA BSC Provides a sterile workspace with laminar airflow and UV sterilization but lacks positive pressure, HEPA filtration to remove airborne DNA, and physical separation of pre- and post-PCR areas. BSCs are designed for biosafety, not aDNA-specific contamination control. aDNA Clean Room Features positive pressure to prevent external DNA entry, HEPA-filtered air, UV-irradiated surfaces, and segregated pre- and post-PCR facilities to prevent amplicon carryover. These are essential for aDNA to avoid modern DNA contamination, which is abundant in microbial-rich environments like paleofeces. The BSC’s limitations likely contributed to the 16S rRNA contamination, as environmental DNA (e.g., from lab air, surfaces) could have entered during grinding or transfer. This is critical for aDNA, where low-yield, degraded samples are easily overwhelmed by modern DNA.

- The study does not report removing the outer layer of paleofeces samples before extraction, a standard practice in aDNA studies to eliminate surface contaminants. Paleofeces from archaeological contexts are exposed to environmental DNA (e.g., soil microbes, animal feces) during burial and excavation. The outer layer, especially in a mixed midden (Ref 28), is likely contaminated with modern or non-target DNA. This could lead to false positives or misattribution of microbial signals.

- The study does not mention applying UV irradiation to paleofeces samples prior to DNA extraction, a critical step in aDNA protocols to degrade modern surface DNA contamination. UV irradiation (e.g., 254 nm, 30–60 minutes) is commonly used in aDNA studies to cross-link and degrade modern DNA on sample surfaces, reducing contamination risks from handling, excavation, or environmental exposure. Paleofeces, being organic and porous, are particularly prone to modern microbial DNA adhesion.

- The study relies on real-time PCR (TAC) and digital PCR fluorescence signals (Cq values, manual thresholding) without sequencing amplicons to confirm their identity. Sequencing (e.g., Sanger or next-generation sequencing) is essential in aDNA studies to verify that amplified products are ancient, not modern contaminants or non-specific amplifications.

- The primers and extraction method were validated on modern feces (Ref 34), not aDNA, which is fragmented and chemically modified (e.g., deamination). This risks false negatives or reduced sensitivity.

- The absence of photographic documentation of the paleofeces (coprolites) in the study is a significant oversight, as visual evidence of their appearance could have provided critical clues about their probable origin (human vs. canine) based on shape, size, texture, or dietary inclusions, especially in the mixed human-canine Rio Zape midden (Ref 28)

**Do you want your identity to be public for this peer review?** For information about this choice, including consent withdrawal, please see our Privacy Policy

Reviewer #1: No

Reviewer #2: No

Reviewer #3: No

Reviewer #4: No

---

## [Author Response · Author response to Decision Letter 2]

27 Jul 2025

We have attached our response to the reviewers.

---

## [Decision Letter · Decision Letter 2]

28 Aug 2025

Dear Dr. Capone,

Thank you for submitting your manuscript to PLOS ONE. After careful consideration, we feel that it has merit but does not fully meet PLOS ONE’s publication criteria as it currently stands. Therefore, we invite you to submit a revised version of the manuscript that addresses the points raised during the review process.

We look forward to receiving your revised manuscript.

Kind regards,

Elham Kazemirad, Ph.D

Academic Editor

PLOS ONE

Journal Requirements:

Reviewers' comments:

Reviewer's Responses to Questions

**Comments to the Author**

Reviewer #3: (No Response)

2. Is the manuscript technically sound, and do the data support the conclusions?

Reviewer #3: Partly

3. Has the statistical analysis been performed appropriately and rigorously?

Reviewer #3: No

4. Have the authors made all data underlying the findings in their manuscript fully available?

Reviewer #3: No

5. Is the manuscript presented in an intelligible fashion and written in standard English?

Reviewer #3: No

Reviewer #3: Here’s the thing: the study is interesting and shows proof-of-concept that targeted qPCR can recover enteric pathogen–associated gene signals from paleofeces. However, as currently written the manuscript over-claims (prevalence, “first detections”) and lacks the experimental and data transparency required to support those organismal and epidemiological conclusions. Major revision is required. Below I list the problems that must be fixed and the concrete actions I would insist on before recommending acceptance.

Major issues (must be addressed)

Sequence confirmation of positives

qPCR signal alone is insufficient to claim organismal detection or novelty. For all key positive assays (those used to claim “first detection” or prevalence), the authors must provide sequence confirmation (Sanger or amplicon sequencing) of the amplified products and deposit sequences (GenBank accessions). If sequencing is not possible, all organismal/novelty claims must be downgraded to “qPCR signal consistent with X — sequence confirmation not obtained.”

Transparent replicate-calling policy and raw data

Provide a clear “Calling policy and concordance” subsection in Methods that states how positives were called (e.g., Cq threshold, replicate rule).

Supply a supplementary table with per-replicate raw Cq/Ct values for every assay, sample, extraction blank, and no-template control.

In the main results (or Table 1) show prevalence under both a conservative rule (e.g., ≥2/3 replicates positive) and a liberal rule (any replicate positive). This lets readers judge robustness.

Negative controls and contamination handling

Present extraction/reagent blank Cq values in full.

Any target detected in negative controls should be excluded from prevalence claims unless sequence-confirmed and shown to differ from reagent contaminants. Describe how control results modified calling.

Host identification

Human mtDNA was detected in only 1/10 samples. Either test for common alternative vertebrate mtDNA markers to exclude nonhuman origin for mtDNA-negative samples or explicitly flag those samples as host-uncertain and remove species-specific interpretation from them.

Overstated interpretation and “first detection” claims

Reword statements that imply population-level prevalence or definitive ancient infection. Replace “first detection” with provisional language unless sequence-confirmed. Example for Abstract: “These results provide proof-of-concept that targeted qPCR can recover enteric pathogen-associated gene signals from paleofeces; prevalence estimates are exploratory and constrained by sample size and taphonomic uncertainty.”

Reproducibility / independent validation (preferred)

For the most novel/important detections, independent replication in a second laboratory or with a second method (amplicon sequencing, different primer set) is strongly recommended.

Minor / editorial issues (to be fixed)

Correct spelling errors and formatting artifacts (e.g., “dessicated” → desiccated; remove Word track-change remnants and “Formatted:” markers).

Standardize and italicize Latin names (Blastocystis spp., Entamoeba histolytica, etc.) and acronyms (define qPCR, Cq on first use).

De-duplicate and correct references (several duplicated entries noted).

Ensure figure and table legends are self-contained and that all acronyms used in tables are defined.

Provide per-table legends for all supplementary data files.

Data availability (required for PLOS policy)

Deposit sequence data (Sanger/amplicon reads) to GenBank (provide accession numbers) and any raw sequencing reads to SRA if applicable.

Provide downloadable CSV/TSV files with all per-replicate qPCR raw Cq/Ct values, negative-control Cq values, sample metadata (sample ID, context/depth/dating, extraction batch, lot numbers where relevant), and the exact calling-policy script or spreadsheet. Upload these to a public repository (Dryad, Zenodo, figshare) and link them in the Data Availability Statement.

Suggested concrete edits you can paste

Abstract (replace opening): “These results provide proof-of-concept that targeted qPCR can recover enteric pathogen-associated gene signals from paleofeces; prevalence estimates are exploratory and constrained by sample size and taphonomic uncertainty.”

Methods: add subsection “Calling policy and concordance” detailing Cq thresholds and replicate rules.

Results/Table 1: add columns “Conservative (≥2/3 positives)” and “Liberal (any positive)”.

Ethical / publication concerns

I found no evidence of dual publication or ethical misconduct beyond the contamination/control concerns already raised. The main issue is analytic rigor and transparency; once the data and sequences are provided and controls addressed, ethical concerns will be resolved.

**Do you want your identity to be public for this peer review?** For information about this choice, including consent withdrawal, please see our Privacy Policy

Reviewer #3: No

---

## [Author Response · Author response to Decision Letter 3]

12 Sep 2025

Editor at PLOS ONE,

Please see our cover letter for our response to the request for revision.

Regards,

Drew Capone

---

## [Editor Report · Decision Letter 3]

16 Sep 2025

Targeted pathogen profiling of ancient feces reveals common enteric infections in the Rio Zape Valley, 725-920 CE

PONE-D-25-01954R3

Dear Dr. Capone,

We’re pleased to inform you that your manuscript has been judged scientifically suitable for publication and will be formally accepted for publication once it meets all outstanding technical requirements.

Kind regards,

Elham Kazemirad, Ph.D

Academic Editor

PLOS ONE
---

## [Editor Report · Acceptance letter]

PONE-D-25-01954R3

PLOS ONE

Dear Dr. Capone,

I'm pleased to inform you that your manuscript has been deemed suitable for publication in PLOS ONE. Congratulations! Your manuscript is now being handed over to our production team.

Kind regards,

on behalf of

Dr. Elham Kazemirad

Academic Editor

PLOS ONE